# Retrospective Survey of Dog and Cat Endoparasites in Ireland: Antigen Detection

**DOI:** 10.3390/ani13010137

**Published:** 2022-12-29

**Authors:** Theo de Waal, Sandra Aungier, Amanda Lawlor, Troy Goddu, Matthew Jones, Donald Szlosek

**Affiliations:** 1School of Veterinary Medicine, University College Dublin, D04 W6F6 Dublin, Ireland; 2IDEXX Laboratories Inc., Westbrook, ME 04092, USA; 3IDEXX Laboratories Ltd., Wetherby LS22 7DN, UK

**Keywords:** endoparasites, dogs, cats, antigen detection, zoonosis, Ireland

## Abstract

**Simple Summary:**

Untreated and stray dogs and cats, in particular, play an important role in contaminating the environment with important zoonotic parasites. Archive faecal samples collected between 2016–2019 from dogs (n = 789) and cats (n = 241) were examined using the IDEXX Fecal Dx^TM^ and SNAP^TM^ *Giardia* antigen assays for the detection of *Toxocara*, hookworms, *Trichuris* and *Giardia* infections. *Giardia duodenalis* was the most common parasite (26%) detected in the dogs, followed by ascarids (17.6%) and hookworms (5.3%). *Trichuris vulpis* was only detected in 1 dog. Ascarids (23.2%) was the most common parasite detected in the cats, followed by *Giardia* (12.9%) and hookworms (2.9%). This study shows a high prevalence of parasite infection in untreated and stray dogs and cats in the greater Dublin area in Ireland. Since they live in synanthropic conditions and can roam over vast distances, they can contaminate public areas and pose a risk to both humans and owned pets that utilise these spaces. It is therefore important to raise public awareness and increase the knowledge on zoonotic parasites.

**Abstract:**

Endoparasites of dogs and cats, play an important role in both veterinary medicine and public health. Untreated and stray dogs and cats, in particular, play an important role in contaminating the environment with important zoonotic parasites. Thus, the aim of this study was to estimate the prevalence of intestinal parasites in stray dogs and cats using highly sensitive and specific copro-antigen tests. Archive faecal samples from previous surveys conducted between 2016–2019 from dogs (n = 789) and cats (n = 241) were included in this study. The IDEXX Fecal Dx™ antigen panel was used for the detection of *Toxocara*, hookworms, *Trichuris* and the SNAP™ *Giardia* antigen assay was used for the detection of *Giardia* infection. *Giardia duodenalis* was the most common parasite (26%, n = 205) detected in the dogs, followed by ascarids (17.6%, n = 139) and hookworms (5.3%, n = 42). *Trichuris vulpis* was only detected in 1 dog. Ascarids (23.2%, n = 56) was the most common parasite detected in the cats, followed by *Giardia* (12.9%, n = 31) and hookworms (n = 7, 2.9%). No whipworms were detected in cats. Overall, there was little difference in the positivity between sexes in both dogs and cats. However, in terms of age, adolescent dogs (<3 years) and kittens (<1 year) had the highest parasite prevalence overall, with *G. duodenalis* and ascarids being the most prevalent. This study shows a high prevalence of parasite infection in untreated and stray dogs and cats in the greater Dublin area in Ireland. Since they live in synanthropic conditions and can roam over vast distances they can contaminate public areas and pose a risk to both humans and owned pets that utilise these spaces. It is therefore important to raise public awareness and increase the knowledge on zoonotic parasites.

## 1. Introduction

Dogs and cats are host to a number of gastrointestinal parasites. Apart from the direct disease impact on the infected animal, some of them are also of zoonotic importance [1,2]. In 2020, the population of pet cats in Europe was over 110 million and dogs over 90 million (https://www.statista.com/statistics/453880/pet-population-europe-by-animal/ (accessed on 28 November 2022)). In 2019, about one in five households in Europe owned at least one pet dog and there is a general increase in the number of pets over the last decade. In Ireland 25% of households owned at least one pet dog and 17% at least one pet cat. Pet ownership in Ireland appears to have increased over the last decade (https://www.statista.com/statistics/517020/households-owning-cats-dogs-europe-ireland/ (accessed 28 November 2022)); dogs: 425,000–455,000; cats: 310,000–325,000, from 2010 to 2020. The increase in owned dog/cat populations has also led to an increase in stray dog and cat populations as well as the continual issue of animal faecal material in public places. There are several campaigns to create awareness and the potential health implication (https://www.dogstrust.ie/whats-happening/news/the-big-scoop-2020; https://www.nytimes.com/2018/01/16/health/toxocara-children-new-york-playgrounds.html (accessed on 28 November 2022)). The number of stray dogs in Ireland is estimated to be 2640, per million people (https://www.petethevet.com/the-stray-dog-issue-overseas-and-in-ireland-podcast-from-pete-the-vet-on-newstalks-pat-kenny-show/ (accessed on 28 November 2022)). According to the International Companion Animal Management Coalition (ICAM) (https://www.icam-coalition.org/download/humane-dog-population-management-guidance/ (accessed on 28 November 2022)), members of the public and government authorities are concerned about public health and safety issues associated with stray animals.

Two important zoonotic gastrointestinal nematodes of dogs and cats are *Toxocara canis* and *T. cati*, respectively. Humans may become infected with these parasites, where the larvae migrate and encyst in tissues and organs, surviving for months or sometimes even years. Although the vast majority of human *Toxocara* infections are asymptomatic [3], infection in humans may cause several clinical syndromes described as visceral larva migrans (VLM), ocular larva migrans (OLM), covert toxocarosis and neural larva migrans (NLM) [4,5,6]. Humans mainly become infected through the ingestion of embryonated eggs and therefore public parks and playground sandpits contaminated with dog or cat faeces play a crucial role in the perpetuation of this infection. In Europe high levels of environmental contamination with *Toxocara* spp. have been reported [7,8,9,10,11]. A study in Portugal found 85.7% of sandpits and 50% of parks examined were contaminated with Toxocara spp. eggs, while 85.5% of the sandpits and 34.4% of the parks were contaminated with *Toxocara cati* eggs [11]. The high prevalence in sandpits is in agreement with other studies [12], whereas *T. canis* is mostly found in soil in public parks. In a recent survey of parks in the greater Dublin area, Ireland, 5/8 (63%) parks were positive for *Toxocara* spp. eggs, with the prevalence of soil contamination ranging between 0 and 13.7% (Keegan 2022, unpublished data).

Another potential zoonotic parasite is *Giardia duodenalis*, which is widely distributed and can infect multiple hosts, including dogs, cats and humans. Some studies have shown that the risk of gastrointestinal parasitism in dogs (including *G. duodenalis*) may increase with park use in urban areas [13] suggesting that recreational behaviours in parks and certain demographics are risk factors for parasitism in pet dogs. Recent studies in Europe [14,15,16,17,18] indicate the prevalence of *G. duodenalis* ranging between 20% and 57% in dogs. Fewer studies have been done in cats, a recent meta-analysis found that 2.3% of cats tested positive for *G. duodenalis*, globally. European studies have found prevalence rates ranging from 5.9% to 20.5% in cats [19,20].

Recent molecular studies on *G. duodenalis* have shown it to be a multispecies complex consisting of eight distinct genotypes (assemblages), some of which are host specific, while others are regarded as potentially zoonotic [21,22,23]. There is epidemiological and molecular evidence that supports the zoonotic transmission of *G. duodenalis* among humans and dogs living in the same community showing a highly significant association between the prevalence of *G. duodenalis* in humans and presence of a *Giardia*-positive dog in the same household (odds ratio 3.01, 95% CI, 1.11, 8.39, *p* < 0.001) [24,25]. *Giardia* is a notifiable disease in humans in Ireland and recent data (https://www.hpsc.ie/abouthpsc/annualreports/ (accessed on 28 November 2022)) indicated a crude incidence rate (CIR) of 5.7 per 100,000 population in 2018 which is an increase of 13% in the CIR compared to 2017. Overall, there has been a noticeable increasing trend in the number of notifications reported over the last five years (145 in 2015 to 252 in 2019). Infections are thought to be mostly water-borne but specific molecular typing was not performed so it is not possible to say if any of these infections were from zoonotic genotypes (assemblages).

In conventional parasitology, parasitic elements are detected using a variety of flotation techniques [26]. In many parasitic infections and in ascarids such as *Toxocara*, in particular, a negative correlation has been found between number of eggs detected and the number of adult worms in the host [27]. These techniques have limited sensitivity and are influenced by a number of factors such as the variations in the number of eggs/cysts produced by the different parasites at different stages of its life cycle, the adequacy of the sample (e.g., sample size, freshness, lack of contamination with free-living organisms), the flotation technique (e.g., passive flotation, double centrifugation), the flotation solution and the experience of the person reading the slide to correctly identify the parasite.

For this reason, recent research has focused on alternative, more sensitive, methods to detect parasite infection in dogs and cats. Molecular or immunological methods to detect helminth stages or their antigens are alternative assays that can be both specific and sensitive to diagnose infections [28]. Immunological assays that detect specific parasite antigens (secreted/excreted (ES) proteins) released in the host faeces (coproantigens) are important tools for screening and surveillance programmes. The detection of ES proteins is advantageous because these antigens are present throughout the life cycle of the parasite, including late in the prepatent period when eggs are not yet produced. Detection of antigen requires a much smaller amount of faecal material and therefore may not be negatively impacted, as are flotation assays, by small sample amounts. The detection of coproantigens of protozoal infections of *Giardia* and *Cryptosporidium*, has been routinely used in veterinary medicine for many years [29,30]. More recently three coproantigen ELISA have recently been developed and validated for dog and cat nematodes; *Trichuris vulpis* [31], *Ancylostoma caninum, Toxocara canis* and *T. cati* [32]. These tests have been validated in both naturally and experimentally infected animals and have been shown to have a high sensitivity and specificity [31,32] with coproantigen of *A. caninum* detected 2 weeks before the patency period or in the case of *T. vulpis* approximately 6.5 weeks before the patency period [31,32]. In the case of *T. canis* antigens were detected in 4 out of 5 dogs, 1 week prior to egg shedding. In some cases, the ELISA results closely mimicked the faecal egg counts after anthelmintic treatment. However, in others the coproantigen levels rose again a short time after treatment, which may suggest the establishment of infections from somatic larval stages It has also been shown that these tests are able to detect *Ancylostoma tubaeforme* and *T. cati* in cats [32,33].

The aim of this study was to assess parasite prevalence in stray dogs and cats in the greater Dublin area, Ireland using the more sensitive coproantigen ELISA.

## 2. Materials and Methods

A total of 1030 archive faecal samples from dogs (n = 789) and cats (n = 241) from previous surveys in Ireland, between 2016–2019, were used in this study. The samples were collected from unowned dogs and cats from animal shelters and were sampled when animals entered the shelter, before routine anthelmintic treatment. Where possible, the breed, age and sex of the animals sampled were recorded as well as faecal consistency (diarrhoeic [watery, no texture], loose [moist, some texture, leaves residue when manipulated] or formed [hard, dry, firm and holds form when manipulated]). From each sample originally submitted, approximately 1 g of faeces was transferred to an Eppendorf tube and stored at −20 °C. These archived samples were then used in the coproantigen assay to detect ascarids, hookworms and whipworms using the commercial Fecal Dx™ antigen assay (IDEXX Laboratories, Inc., Westbrook, ME, USA). In addition, *Giardia* antigen was also detected in the samples using the SNAP™ Giardia test (IDEXX Laboratories, Inc., Westbrook, ME, USA). All the coproantigen assays were performed at IDEXX laboratories in Wetherby, UK according to the manufacturer’s protocol.

### Statistical Analysis

Categorical variables were presented as percent and frequency. Canine life stages were defined as follows: Adolescent (<3 years), Adult (>=3 years and <7 years), Senior (>=7 years and <11 years), Geriatric (>=11 years). For felines, life stages were defined as follows: Adolescent (<1 years), Adult (>=1 years and <8 years), Senior (>=8 years and <13 years), Geriatric (>=13 years). Faecal consistency was categorised in three ordered levels, Formed, Loose or Diarrhoea, when information was available. Statistical analysis was done using R version 4.0.2.

## 3. Results

### 3.1. Dogs

Of the 789 dog samples analysed 37.5% were female, 53.2% were male and 14.1% were unknown. The most common dog breed was mixed/other (54.4%, n = 429), followed by Jack Russel Terrier (11.9%, n = 94), Terrier (7.4%, n = 58), and Collie (2.7%, n = 21). The majority (50.4%, n = 398) of dogs were <3 years of age, the age profile of the dogs is summarised in Table 1.

*G. duodenalis* was the most common parasite (26%, n = 205) detected in the dogs, while *Trichuris vulpis* was only detected in 1 dog. The prevalence of infection of the different pathogens detected are summarised in Figure 1.

The prevalence of the different parasites detected in the different life stages is given in Table 2.

There was no significant differences in overall positivity between the sexes. However, when looking at the individual parasites it appears to be multidirectional with *G. duodenalis* and ascarids being more prevalent in male than female dogs. Adolescent dogs were observed to have the highest parasite burden (Table 2) overall, with *G. duodenalis* and ascarids being the most prevalent. Interestingly, whipworms were only detected in an adolescent dog, with *G. duodenalis* being the most prevalent pathogen detected in all age groups.

The faecal consistency of the majority of samples analysed was formed faeces (81.4%, n = 642) and no diarrhoeic samples was analysed. There was no clear relationship between faecal consistency and parasite infection (Figure 2).

### 3.2. Cats

Of the 241 cat samples analysed 46.5% were female, 42.3% male and 11.2% were unknown. The majority (25.7%, n = 62) of the cats were <1 year of age, and the age profile of the cats is summarised in Table 3.

Ascarids were the most common parasite (23.2%, n = 56) detected in the cats, while no whipworms were detected in any of the samples. Overall, the prevalence of infection of the different pathogens detected is summarised in Figure 3.

The prevalence of the different parasites detected in the different life stages is given in Table 4.

As was the case with dogs there was little difference in overall positivity between the sexes. However, when looking at the individual parasites, more male cats (19.4%) were positive for *G. duodenalis* than female cats (6.2%) and ascarid infection was similar (23%) between male and female cats.

Adolescent cats were observed to have the highest parasite burden (Table 4) overall, with *G. duodenalis* and hookworms being the most prevalent. Interestingly, no whipworms were detected in any of the samples, with hookworms being the most prevalent pathogen detected in all age groups.

The faecal consistency of the majority of samples analysed was from formed faeces (94.2%, n = 227) and only 3 (1.2%) of the samples were classified as diarrhoeic. *G. duodenalis* was mostly found in diarrhoeic samples while ascarid and hookworms were only found in formed faecal samples (Figure 4).

## 4. Discussion

Endoparasites were frequently found in stray dog and cat samples in this study, with 49% of dogs and 39% of cats harbouring one or more parasite infections. This is in agreement with a recent study in Dublin, Ireland that 30% of stray dogs and 41% of stray cats were harbouring at least one parasite infection [34]. In fact, similar findings have been found worldwide [35,36,37,38,39,40,41,42,43,44,45,46,47,48,49,50].

In the present study, there was little difference in the positivity between the sexes in both dogs and cats. However, in terms of age, adolescent dogs and cats had the highest parasite prevalence overall, with *G. duodenalis* and ascarids being the most prevalent (Table 2 and Table 4). This again underlies the findings of previous studies that a free-living lifestyle and a young age are major risk factors for endoparasite infections [48,51]. The result from the present survey also highlights the fact that stray dogs and cats frequently harbour zoonotic parasites that are of key veterinary and public health concerns in line with other surveys [52]. None of the cats and only 1 dog tested positive on the coproantigen test for *Trichuris* spp. Most recent studies on the prevalence in cats in North America, Europe and Australia have shown a very low prevalence [53] with the exception of St Kitts where the prevalence of infections among non-owned/feral cats was 71% [54]. However, the low prevalence in dogs is surprising since *T. vulpis* eggs are known to survive for long periods in the environment especially in temperate climates [55] where it is a constant source of infection and often leads to high infection rates in dogs. Reports from Canada, Belgium and Holland [56,57,58] have found that *T. vulpis* is the second most common helminth. Older dogs tend to be more commonly infected with *T. vulpis* [59], but interestingly in this study the positive dog was a young dog (<3 years of age). A survey in Poland found 8.4% of dogs were infected with *T. vulpis* [50]. In Hungary 13–48% and Italy 10–18% of dogs were infected with *T. vulpis* [50,51,52,53,54,55,56,57,58,59,60,61]. The zoonotic importance of *T. vulpis* is still controversial, as there are only a few case reports in humans [55,62] but diagnosis was only based on egg measurement.

*Giardia* was the most frequent intestinal parasite found in dogs (26%) in this study. Similar prevalence was also recorded in many other studies, for example 24.8% of dogs in a large study in Europe was positive for *G. duodenalis* [63], 22.7% in Belgium [64] and 21.0% in the UK [16]. A few surveys recorded a much higher prevalence, with 59% of dogs in Hungary [65], 55% in Italy [66] and 63% in Spain [67] infected with *G. duodenalis*. In cats, *G. duodenalis* was the second most frequent intestinal parasite found (13%) which is considerably lower than that found in a multi-county study in Europe where the infection rates in cats were 20.3% [63] but in line with the pooled prevalence (12%) calculated in a worldwide meta-analysis [20]. Differences in the sensitivity and specificity of the diagnostic assays to detect *G. duodenalis* infection may contribute to the large differences in *Giardia* prevalence. It has also been suggested that the risk of *G. duodenalis* infection is increased with the increased frequency of anthelmintic administration vacating a niche in the intestine by the removal of other parasites [68]. However, it is unlikely to have played a role in this survey as many dogs and cats were infected with more than one parasite and also were unlikely to have been subjected to frequent anthelmintic treatments. Interestingly we have noticed an increasing incidence in positive *G. duodenalis* cases in dog samples submitted to the University College Dublin, Veterinary Hospital Parasitology laboratory over the last 10 years (Lawlor, 2022 unpublished data). The clinical significance is unclear as most infections in immunocompetent animals are asymptomatic, but it may be an important zoonosis. *G. duodenalis* is regarded as one of the 5 most important parasitic diseases of humans in Europe [69]. However, most molecular studies have reported that cats and dogs are mostly (but not exclusively) infected with host-specific parasites [25,70]. An old study [71] on the risk factors for people becoming infected with *G. duodenalis* through exposure to companion animals, found that dog, cat or farm animal contact was not a risk factor for giardiosis in humans, while a more recent study from the UK [16] found only the host-specific *G. duodenalis* assemblages C and D in rescue shelter dogs. A recent review by Cai et al. [25] concluded that the zoonotic transmission of *G. duodenalis* is probably less common than believed and that it could perhaps mainly be attributed to the contamination or contact with just a few species of animals such as rabbits, guinea pigs, equines, nonhuman primates, chinchillas, and beavers.

*Toxocara* spp. are the most common ascarid of dogs and cats, and in this study, it was also the most frequent helminth detected in both dogs (139/789; 17.6%) and cats (56/241; 23.2%). Infection rates were also higher in the younger cohort (26% & 32%, respectively). These results are in general agreement with many other studies where prevalence ranged in cats and dogs from 8–76% [72] and 1–30% [49,73] in Europe, respectively, and to previous results in Ireland of 16% and 30% in dogs and cats, respectively [34]. The higher prevalence in cats may be attributable to the route of transmission (lactogenic), different environments and food supply and eating behaviour of cats [74]. Human toxocarosis is the most common parasitic infection from pets [75] so it would be important that proper biosecurity involving disinfection and cleaning schedule be followed in animal rescue centers. ESCCAP emphasises the picking up and disposal of animal faeces (https://www.esccap.org/page/GL6+Control+of+Intestinal+Protozoa+in+Dogs+and+Cats/30/ (accessed on 28 November 2022)) and good hygiene after playing with animals as important preventative measures.

The only hookworm in Ireland is *Uncinaria stenocephala*, and prevalence of infection in this study was generally low with only 5% of dogs and 3% of cats infected (Figure 1 and Figure 3), similar to the previous findings in Ireland (5% of dogs, 0% cats) [34] and Germany (1% in dogs) [49].

In general, the coproantigen ELISA have a greater sensitivity for the detection of parasites compared to conventional centrifugal-flotation techniques [76] and can detect nematode antigens before infections become patent [31,32]. This is useful as a surveillance test where parasite egg levels may be very low.

## 5. Conclusions

In conclusion, a weakness of the study was that the previous history of parasite control of the animals was unknown or for how long they have been homeless as this may have had an influence on the parasite prevalence recorded here. Nevertheless, this study provides updated information on the prevalence of parasite infections in a population of unwanted and stray dogs and cats in the greater Dublin area in Ireland, confirming a high prevalence of parasite infection. Since they are not subjected to regular anthelmintic treatment, live in synanthropic conditions and can roam over vast distances they can contaminate public areas, and pose a risk to both humans and owned pets that utilise these spaces [77,78,79]. It is therefore important that animal shelters adopt good hygiene and sanitary approaches in order to prevent contamination of the environment, to protect workers against zoonotic pathogens, to raise public awareness and to increase knowledge on zoonotic parasites.

## Figures and Tables

**Figure 1 animals-13-00137-f001:**
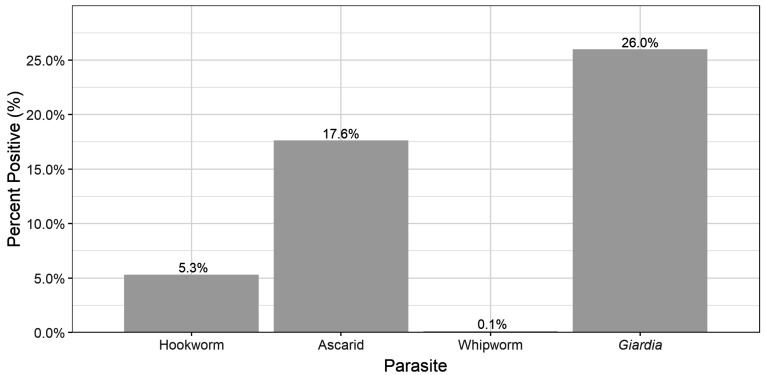
Prevalence of parasites detected in dogs.

**Figure 2 animals-13-00137-f002:**
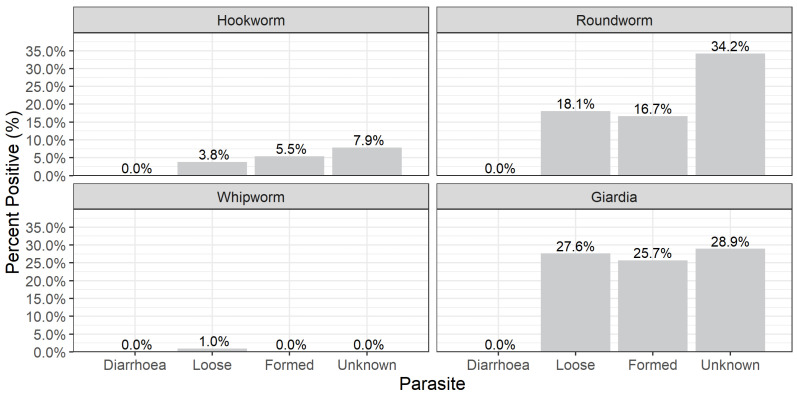
The relationship between faecal consistency and parasites detected in dog samples.

**Figure 3 animals-13-00137-f003:**
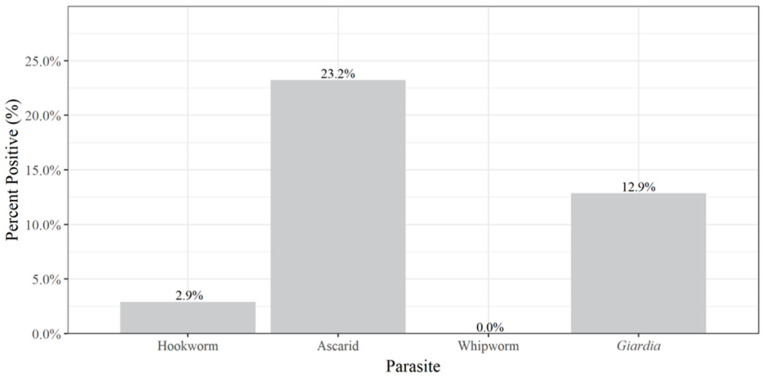
Prevalence of parasites detected in cats.

**Figure 4 animals-13-00137-f004:**
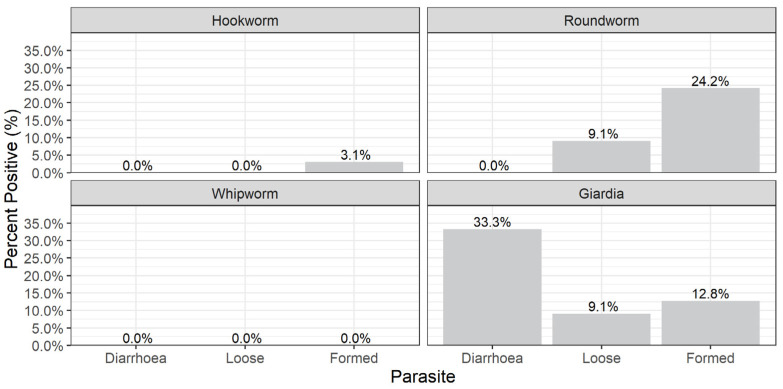
The relationship between faecal consistency and parasites detected in cat samples.

**Table 1 animals-13-00137-t001:** Age profile of dogs (n = 789) examined for endoparasites.

Category	n	%
Unknown	171	21.7%
Adolescent	398	50.4%
Adult	160	20.3%
Senior	46	5.8%
Geriatric	14	1.8%

**Table 2 animals-13-00137-t002:** Prevalence of parasites detected in the different life stages of dogs.

Parasite	Adolescent (n = 398)		Adult (n = 160)		Senior (n = 46)		Geriatric (n = 14)		Unknown (n = 171)	
	n	%	n	%	n	%	n	%	n	%
Giardia	125	31.4	32	20.0	12	26.1	3	21.4	33	19.3
Hookworm	26	6.5	8	5.0	4	8.7	0	-	4	2.3
Ascarid	105	26.4	17	10.6	1	2.2	0	-	16	9.4
Whipworm	1	0.3	0	-	0	-	0	-	0	-

**Table 3 animals-13-00137-t003:** Life stages of cats (n = 241) examined for endoparasites.

Category	n	%
Unknown	55	22.8
Adolescent	62	25.7
Adult	114	47.3
Senior	8	3.3
Geriatric	2	0.8

**Table 4 animals-13-00137-t004:** Prevalence of parasites detected in the different age categories of cats.

Parasite	Adolescent (n = 62)	%	Adult (n = 114)	%	Senior (n = 8)	%	Geriatric (n = 2)	%	Unknown (n = 55)	%
	**n**		**n**		**n**		**n**		**n**	
Giardia	13	21.0	13	11.4	1	12.5	0	-	4	7.3
Ascarid	20	32.3	21	18.4	1	12.5	0	-	14	25.5
Hookworm	3	4.8	4	3.5	0	-	0	-	0	-

## Data Availability

The data presented in this study are available on request from the corresponding author. The data is not publicly available.

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
