# Peer review of "Retrospective Survey of Dog and Cat Endoparasites in Ireland: Antigen Detection"

_animals, 2022, doi:10.3390/ani13010137_

Round 1

Reviewer 1 Report

This manuscript presents on the use of coproantigen to detect parasites in stray dogs/cats, as a tool for surveillance and to understand risks to people and owned dogs/cats. The data are clear and the importance of work like this is well presented. However, the manuscript could use a good read-through for grammar and typos. Some of these are below along with some minor comments.

Simple summary

Lines 12, 24: the first sentence seems to be missing a comma somewhere. Maybe “cats, in particular, play an…”

Lines 13, 25: remove “important” or replace important in line 12 with significant and keep important on line 13

Lines 15, 30: replace Uncinaria with hookworms. If I understand the antigen test correctly, if does not distinguish hookworm genera/species. Within the body of the manuscript, it can be clarified that only Uncinaria occurs in Ireland. However, not knowing the source of the dogs/cats, there is the chance (albeit slim) that the animal could have Ancylostoma, if recently imported when the sample was collected.

Lines 19, 37: unwanted or untreated?

Line 35: maybe change “but” to “however”

Introduction

Lines 53-55: a ) is missing somewhere

Lines 55-65: the transition from owned animals to stray ones is a little unclear and these lines could do with a bit of rewording to make the flow better. Maybe “The increase in owned dog/cat populations has also led to an increase in stray dog and cat populations”. Also, littering and fouling have different meanings depending on the English (Irish, British, American). Maybe reword to defecation or the issue of defecation in public spaces or the issue of animal fecal material in public spaces.

Line 67: deadend host vs paratenic since the life cycle does not continue once a person is infected

Line 70: asymptomatic [3], infection

Line 73: space between %)with

Line 74: remove with before highest

Lines 76-77: I think this might be easier to read if changed to “in South-East Asia; 24.2% (16.0–33.5%) in the Western Pacific; 22.8% (19.7–26.0%) in the Americas; and 10.5% (8.5–12.8%) in Europe

Line 78: I don’t think Irish is needed, since it is in Ireland. Also, add a comma after 31%

Line 80: 2000s

Lines 87 and 302: spp. (add the . after spp)

Lines 92-95: not sure if this last sentence is needed. It doesn’t really relate to the discussion.

Lines 97-100: This sentence might fit better in the first paragraph

Line 138: detection is typically only a few days sooner then when eggs appear, so this statement might need some modification. It might be worth mentioning that single sex infections can be detected.

Lines 138-139: detection of antigen requires a much smaller (remove “only”)

Line 147: this time of detecting during the PPP is highly dependent on the strain and the PPP. In areas with a PPP of only 14 days, detection is about 3-5 days before eggs. This section could be reworded to clarify a bit more.

Line 151: typo form to from

Line 152: remove one of the alsos

Materials and Methods

Line 161: age, and sex (add “and”)

Line 177: remove the ) after diarrhoea

Materials and Methods

Lines 180-182: remove

Tables 1 and 2. include the N of 789 in the title vs listing 5 times for table 1. For table 2, list the N by the age e.g., Adolescent (N=398). Same issues with tables 3 and 4.

Line 191: italics for Giardia and Trichuris vulpis

Line 206: were vs was

Discussion (please review for grammar/typos)

Line 240: infections vs infection

Line 264: that vs the before T. vulpis. On a side note, when looking for T vulpis previously, I found it most frequently in collies and greyhounds. Also, it was highest in July/August. So the lack of seeing it might be due to the breeds and time of year samples were collected.

 References

Please double check formatting, especially for 2, 11, 12, 19, 21, 26, 48, 74, 78, 80 and 84

Author Response

Simple summary
Lines 12, 24: the first sentence seems to be missing a comma somewhere. Maybe “cats, in particular, play an…”
Corrected

Lines 13, 25: remove “important” or replace important in line 12 with significant and keep important on line 13
Corrected

Lines 15, 30: replace Uncinaria with hookworms. If I understand the antigen test correctly, if does not distinguish hookworm genera/species. Within the body of the manuscript, it can be clarified that only Uncinaria occurs in Ireland. However, not knowing the source of the dogs/cats, there is the chance (albeit slim) that the animal could have Ancylostoma, if recently imported when the sample was collected.
Corrected. Although we do not know the history of the dogs, it is very unlikely to be Ancylostoma. Not reported in this paper but all samples were also examined with using classical parasitological techniques and all hookworm infections was identified as Uncinaria. 

Lines 19, 37: unwanted or untreated?
Change to “untreated”

Line 35: maybe change “but” to “however”
Changed to “However,”

Introduction
Lines 53-55: a ) is missing somewhere
Deleted ) in L60

Lines 55-65: the transition from owned animals to stray ones is a little unclear and these lines could do with a bit of rewording to make the flow better. Maybe “The increase in owned dog/cat populations has also led to an increase in stray dog and cat populations”. Also, littering and fouling have different meanings depending on the English (Irish, British, American). Maybe reword to defecation or the issue of defecation in public spaces or the issue of animal fecal material in public spaces.
Corrected as suggested.

Line 67: deadend host vs paratenic since the life cycle does not continue once a person is infected
L73 Changed to “become infected with”

Line 70: asymptomatic [3], infection
Corrected

Line 73: space between %)with
Corrected

Line 74: remove with before highest
Corrected

Lines 76-77: I think this might be easier to read if changed to “in South-East Asia; 24.2% (16.0–33.5%) in the Western Pacific; 22.8% (19.7–26.0%) in the Americas; and 10.5% (8.5–12.8%) in Europe
Corrected

Line 78: I don’t think Irish is needed, since it is in Ireland. Also, add a comma after 31%
Corrected

Line 80: 2000s
Corrected

Lines 87 and 302: spp. (add the . after spp)
Corrected

Lines 92-95: not sure if this last sentence is needed. It doesn’t really relate to the discussion.
Corrected

Lines 97-100: This sentence might fit better in the first paragraph
The sentence was left unchanged

Line 138: detection is typically only a few days sooner then when eggs appear, so this statement might need some modification. It might be worth mentioning that single sex infections can be detected.
Sentence changed.

Lines 138-139: detection of antigen requires a much smaller (remove “only”)
Corrected

Line 147: this time of detecting during the PPP is highly dependent on the strain and the PPP. In areas with a PPP of only 14 days, detection is about 3-5 days before eggs. This section could be reworded to clarify a bit more.
Reworded

Line 151: typo form to from
Corrected

Line 152: remove one of the alsos
Corrected

Materials and Methods
Line 161: age, and sex (add “and”)
Corrected

Line 177: remove the ) after diarrhoea
Corrected

Materials and Methods
Lines 180-182: remove
Removed

Tables 1 and 2. include the N of 789 in the title vs listing 5 times for table 1. For table 2, list the N by the age e.g., Adolescent (N=398). Same issues with tables 3 and 4.
Corrected as suggested

Line 191: italics for Giardia and Trichuris vulpis
Corrected

Line 206: were vs was
Corrected

Discussion (please review for grammar/typos)
The manuscript has been revised and grammatical and topographical errors 

Line 240: infections vs infection
Corrected

Line 264: that vs the before T. vulpis. On a side note, when looking for T vulpis previously, I found it most frequently in collies and greyhounds. Also, it was highest in July/August. So the lack of seeing it might be due to the breeds and time of year samples were collected.

Corrected. Probably also related how these specific breeds are kept.

 References
Please double check formatting, especially for 2, 11, 12, 19, 21, 26, 48, 74, 78, 80 and 84
References updated using the MDPI ACS Endnote style guide.

Reviewer 2 Report

This is an interesting and nice paper reporting data on the prevalence of some important intestinal parasites in stray dogs and cats in Ireland. However, some minor changes are needed before the manuscript can be accepted. 

-The introduction is too long and should be shortened (for example by moving some sentences from the introduction into the discussion or eliminating from the introduction those sentences also present in the discussion)

-Giardia duodenalis (syn. G. intestinalis and G. lamblia) is the only species infecting dogs and cats. Therefore, throughout the manuscript, it would be appropriate to replace Giardia with G. duodenalis.

-at line 302, spp should be replaced with spp.

-the references are not in line with the style requested by Animals.

Author Response

Comments and Suggestions for Authors
This is an interesting and nice paper reporting data on the prevalence of some important intestinal parasites in stray dogs and cats in Ireland. However, some minor changes are needed before the manuscript can be accepted. 

-The introduction is too long and should be shortened (for example by moving some sentences from the introduction into the discussion or eliminating from the introduction those sentences also present in the discussion)
The Introduction has been revised and shortened.

-Giardia duodenalis (syn. G. intestinalis and G. lamblia) is the only species infecting dogs and cats. Therefore, throughout the manuscript, it would be appropriate to replace Giardia with G. duodenalis.
Changed throughout manuscript where appropriate.

-at line 302, spp should be replaced with spp.
Corrected

-the references are not in line with the style requested by Animals.
References updated using the MDPI ACS Endnote style guide.

Reviewer 3 Report

The manuscript does have some relevance with respect to veterinary medicine and public health.  However, there are issues that must be addressed throughout the manuscript. The first issue is the in the abstract and discussion the authors stated that the parasite burden was heavy in animals less than three years in dogs and less than one year in cats. This statement is incorrect as parasite burden has to do with the number of parasites per animal/ sample. What should be stated is the prevalence of endoparasitism was higher in younger groups. 

The second major issue is that the introduction is too long and should be truncated and in text citation should be consistent. 

Author Response

The manuscript does have some relevance with respect to veterinary medicine and public health.  However, there are issues that must be addressed throughout the manuscript. The first issue is the in the abstract and discussion the authors stated that the parasite burden was heavy in animals less than three years in dogs and less than one year in cats. This statement is incorrect as parasite burden has to do with the number of parasites per animal/ sample. What should be stated is the prevalence of endoparasitism was higher in younger groups. 
Corrected

The second major issue is that the introduction is too long and should be truncated and in text citation should be consistent. 
The Introduction has been revised and shortened